# Skeletal Muscle Lipid Droplets and the Athlete’s Paradox

**DOI:** 10.3390/cells8030249

**Published:** 2019-03-15

**Authors:** Xuehan Li, Zemin Li, Minghua Zhao, Yingxi Nie, Pingsheng Liu, Yili Zhu, Xuelin Zhang

**Affiliations:** 1School of Kinesiology and Health, Capital University of Physical Education and Sports, Beijing 100191, China; lixuehan@cupes.edu.cn (X.L.); lizemin@cupes.edu.cn (Z.L.); zhaominghua@cupes.edu.cn (M.Z.); nieyingxi2018@cupes.edu.cn (Y.N.); 2National Laboratory of Biomacromolecules, CAS Center for Excellence in Biomacromolecules, Institute of Biophysics, Chinese Academy of Sciences, Beijing 100101, China; pliu@ibp.ac.cn; 3University of Chinese Academy of Sciences, Beijing 100049, China

**Keywords:** lipid droplet, skeletal muscle, metabolic diseases, athlete’s paradox

## Abstract

The lipid droplet (LD) is an organelle enveloped by a monolayer phospholipid membrane with a core of neutral lipids, which is conserved from bacteria to humans. The available evidence suggests that the LD is essential to maintaining lipid homeostasis in almost all organisms. As a consequence, LDs also play an important role in pathological metabolic processes involving the ectopic storage of neutral lipids, including type 2 diabetes mellitus (T2DM), atherosclerosis, steatosis, and obesity. The degree of insulin resistance in T2DM patients is positively correlated with the size of skeletal muscle LDs. Aerobic exercise can reduce the occurrence and development of various metabolic diseases. However, trained athletes accumulate lipids in their skeletal muscle, and LD size in their muscle tissue is positively correlated with insulin sensitivity. This phenomenon is called the athlete’s paradox. This review will summarize previous studies on the relationship between LDs in skeletal muscle and metabolic diseases and will discuss the paradox at the level of LDs.

## 1. Introduction

The lipid droplet (LD) is an organelle that stores neutral lipids in cells and plays an important role in maintaining lipid homeostasis in almost all organisms [1]. There is abundant experimental evidence that LDs interact with other organelles, and that this is mediated by regulatory proteins and enzymes embedded in their surface. LD proteins are also responsible for regulating the size, shape, and stability of LDs, parameters that are associated with various physiological states [2]. Some of the interactions between LDs and other organelles and aspects of LD dynamics are diagramed in Figure 1.

In the past several decades, there has been a worldwide increase in the incidence of lipid metabolic diseases. As high caloric diets have become more affordable and lifestyles more sedentary, an increasing fraction of the world’s population ingests lipids and other calories above their metabolic needs. A chronic positive energy balance results in elevated blood triacylglycerol (TAG) and free fatty acid (FFA) content. This, in turn, leads to the ectopic storage of neutral lipids in non-adipose tissues, such as skeletal muscle, liver, and heart [3]. Two major human adipose tissues are described: white adipose tissue (WAT) and brown adipose tissue (BAT). WAT is used mainly to store energy, while BAT is used mainly to produce heat [4]. Ectopic lipid storage [5,6] refers to the lipids that cannot be consumed and stored in adipose tissues but are stored in non-adipose tissues, which, in turn, causes lipid metabolism disorder as well as lipid toxicity.

Lipid toxicity is thought to interfere with normal cellular functions in a process called lipid toxicity [7,8]. This is the hypothesis that excessive, atypical storage of lipids in non-adipose tissues influences the metabolic homeostasis of the impacted tissues and organs. Lipid toxicity leads to disturbances in cell signaling and the development of insulin resistance, which, in turn, can result in a series of related diseases including type 2 diabetes (T2DM) and non-alcoholic fatty liver disease (NAFLD) [9].

Due to its great tissue mass and large contribution to metabolic demand, skeletal muscle is a particularly consequential cell type for pathologies of lipid homeostasis [10,11]. Lipids are stored as TAG in LDs within skeletal muscle cells, called intramyocellular lipid (IMCL) [12,13]. To meet energy demands, IMCL can be hydrolyzed into FFAs, which are processed through β-oxidation in mitochondria to generate ATP and heat in skeletal muscle [12,14]. Physical training can induce an increase in IMCL pools, which is reflected in an increased LD number and accompanying morphological changes. Interestingly, while a diet-induced increase in IMCL is associated with insulin resistance, similar IMCL accumulation in response to exercise is not [15]. This apparent contradiction is referred to as is the athlete’s paradox [16]. A resolution of the paradox has yet to be fully achieved.

There have been advances in methodological approaches permitting more accurate measurements of IMCL and LDs in skeletal muscle, including the biochemical extraction of TAG, magnetic resonance spectrometry, histochemical staining with immunofluorescence microscopy, and transmission electron microscopy (TEM) [12,13]. Methods have also been recently established for the isolation of muscle LDs permitting proteomic analysis [17]. These new techniques provide more detailed observation of skeletal muscle LDs, which raises a new perspective for the study of the athlete’s paradox.

## 2. Diabetes Mellitus

Diabetes mellitus (DM) is a complex, chronic metabolic disease with multiple causes. The most obvious feature of the disease is a sustained elevation of blood glucose levels, accompanied by long-term disorders in glucose, lipid, and protein metabolism caused by insufficient insulin secretion or insulin non-responsiveness [18]. The definition of DM contains several criteria including fasting blood glucose above 7.0 mmol/L [19]. If DM is not controlled through proper lifestyle and medical intervention, the sequelae include organ failure and peripheral nerve necrosis [20].

There are three common types of DM. Type 1 diabetes mellitus (T1DM) was previously called insulin-dependent diabetes. This type of autoimmune disease is most common in children and adolescents. Immunological destruction of the pancreatic beta cells results in insufficient insulin secretion to stimulate glucose uptake [21]. T2DM is a type of metabolic disease that is characterized mainly by a reduction in insulin sensitivity (insulin resistance), which in turn decreases insulin-stimulated uptake of blood glucose [22]. The third type is gestational diabetes mellitus (GDM). This type occurs in pregnant women and represents a serious complication of childbirth. The condition is specific to pregnant women who have not previously been diagnosed with diabetes [23].

T2DM is by far the most common form of the disease, accounting for over 90% of cases. The early stages of the disease are marked by increasing insulin resistance in the peripheral tissues including muscle, fat, and liver. At first, a compensatory increase in insulin secretion is able to maintain normal glycemic levels [24,25]. However, over time, the heavy demand leads to beta-cell exhaustion and apoptosis. The resulting decrease in insulin secretion along with increasing insulin resistance leads to a loss in glycemic control and dangerous elevations in blood glucose levels [26,27,28].

At the cellular level, lipid homeostasis includes the balance between the absorption of free fatty acids from the blood and the synthesis and hydrolysis of lipids in cells under normal conditions [29]. The ectopic accumulation of lipids in peripheral tissues can interfere with these cellular functions, disturbing lipid homeostasis at the cellular level. Ultimately, this can lead to diseases including, commonly, T2DM. Due to its increasing prevalence and mounting societal costs, T2DM has become the focus of global attention.

When lipids accumulate beyond the capacity of adipose tissue to efficiently store it, the rate of lipid hydrolysis exceeds that of esterification. This results in a sharp rise in blood FFA levels, which has two downstream consequences. First, it drives insulin resistance through fatty acid receptors on the cell surface. Second, FFAs are absorbed by the peripheral tissues, which further disturbs insulin signaling. Since skeletal muscle accounts for more than 70% of the body’s blood glucose intake [30], the high concentration of free fatty acids in the blood of obese patients has the greatest impact on the insulin response of skeletal muscle [31]. The degree of IMCL is positively correlated with insulin resistance [32,33]. Furthermore, the increase in IMCL in skeletal muscle is accompanied by an accumulation of the metabolic intermediates diacylglycerol (DAG) [34] and neuroamide [35], which have also been linked to insulin resistance. The levels of DAG accumulation in skeletal muscle of endurance training and sedentary obese rats are similar [36]. In addition, the concentration of phosphatidylethanolamine molecular species containing palmitoleate is increased in endurance-trained rats, but the opposite is true in sedentary obese rats. These findings indicate that endurance exercise can affect the lipid composition of skeletal muscle. Unfortunately, the LD lipidome of skeletal muscle with or without endurance exercise remains unknown.

## 3. Skeletal Muscle Lipid Droplets and the Athlete’s Paradox

The adoption of an exercise program can alleviate many chronic diseases [37]. An exercise regimen can accelerate metabolism, improve cardiovascular function, and enhance immunity [38]. Indeed, a short-term exercise intervention was found to alleviate induced insulin resistance but also to increase the expression of genes associated with IMCL synthesis, resulting in the accumulation of TAG in skeletal muscles [39]. Another observational study found that physically trained individuals had increased skeletal muscle TAG relative to lean, sedentary people [16]. This is consistent with the known role of IMCL in meeting the energy demand of skeletal muscle [12]. Other studies have found a positive correlation between IMCL levels and insulin sensitivity in those engaged in aerobic training [40,41]. However, this association is surprising since IMCL has also been positively correlated with the development of metabolic diseases and insulin resistance in the general population [11,42,43]. This is the athlete’s paradox.

It is likely that the assessment of IMCL levels is too crude a measure and that this simple metric belies important biochemical differences between diet-induced and exercise-induced accumulation of TAG in skeletal muscle. A more nuanced analysis is required to resolve the paradox. Improvements in biopsy and imaging techniques for studying muscle are allowing for more detailed analysis of the phenomenon linking LDs and insulin sensitivity in muscle tissue.

## 4. Subcellular Compartmentalization of Lipid Droplets

LDs are found in two distinct locations in muscle, just beneath the cell membrane (subsarcolemmal LDs) and between myofibrils (intermyofibrillar LDs) [44]. The total cellular volume of intermyofibrillar LDs greatly exceeds that of the subsarcolemmal pool [44]. There is a substantial drop in total IMCL following acute exercise suggesting the use of IMCL as an energy reservoir for the muscle [11]. Multiple methodologies including stable isotope tracing, magnetic resonance spectroscopy, and fluorescence and electron microscopy support that conclusion [12]. There is evidence that LDs in both subcellular locations contribute to some degree to the energy needs of muscle tissue [11]. However, the pools appear to be biochemically distinct.

An analysis of LDs in muscle tissue before and after exhaustive exercise found a measurable drop in the cellular fraction of the intermyofibrillar but not the subsarcolemmal LDs [44]. Similarly, in another study, a reduction in the intermyofibrillar lipid pools was seen in endurance athletes after moderate and high-intensity exercise [45]. Thus, it is primarily the intermyofibrillar LDs that contribute energy to muscles during periods of high demand. Furthermore, LDs are more abundant in fast-twitch, type I fibers than type II fibers and this distribution is more pronounced in trained athletes [44,46]. LDs in the muscle of trained athletes are relatively small, are predominantly intermyofibrillar, and are more prevalent in fast-twitch, type I fibers. In contrast, T2DM patients accumulate larger, subsarcolemmal LDs that are more abund ant in type II fibers [45]. A measure of total IMCL does not distinguish between these pools, which likely underlies the athlete’s paradox.

## 5. Lipid Droplets and Mitochondria

LDs interact with mitochondria, and the distribution of LDs and mitochondria in skeletal muscle type I muscle fibers have been observed by traditional light microscope and laser confocal three-dimensional reconstruction technique. It was found that LDs are mainly distributed in the aggregation of mitochondria [47]. Meanwhile, peridroplet mitochondria (PDM) in brown adipocytes support LD expansion because Perilipin-5 induces mitochondrial recruitment to LDs and increases the synthesis of TAG dependent on ATP synthase [48].

It has also been reported that the protein content of each intermyofibrillar mitochondrion in rat skeletal muscle is nearly twice as high as that of Subsarcolemmal mitochondria, which means that intermyofibrillar mitochondria have higher activity [49]. At the protein level, SNAP23 is one of the proteins found to regulate the interaction between LDs and mitochondria [50]. In skeletal muscle, SNAP23 is partially localized on the cell membrane and is involved in the translocation of insulin-sensitive glucose transporter 4 (GLUT4) to the cell membrane [51]. In fatty acid treated cells with increased LDs, SNAP23 is more localized on the surface of LDs, which enhances the interaction between LDs and mitochondria and reduces the plasma membrane GLUT4, which, in turn, decreases glucose uptake [51].

Multiple lines of evidence have established that LDs can physically associate with mitochondria, suggesting a functional link in the mobilization and use of energy reserves [40]. In addition to the accumulation of intramyofibrillar LDs, long-term endurance training also increases the biogenesis of mitochondria [52]. There is evidence that mitochondrial dysfunction can lead to insulin resistance [53]. The activity of the electron transfer chain in submembranous mitochondria of patients with type 2 diabetes and obesity is significantly lower than that of thin volunteers. It suggests that mitochondrial dysfunction may also lead to type 2 diabetes. Therefore, a more detailed understanding of the role of IMCL in health and disease will require the techniques of cell biology and biochemistry.

## 6. New Approaches to the Study of Muscle Physiology

The traditional techniques of total lipid extraction and microscopy of biopsies are complemented by LD isolation and spectroscopy. It remains technically difficult for many laboratories to purify LDs from muscle tissue. Our group established a method to isolate LDs as early as 2004 [54] and then improved the method [55]. Using this technique, we have successfully isolated LDs from mouse skeletal muscle and found a close association between LDs and mitochondria. The isolation method provides a means to carry out morphological, biochemical, and functional analyses of muscle LDs [56]. This technique may help illuminate the role of LD contact with mitochondria or other organelles in health and disease along with molecular details of muscle LD physiology.

The application of magnetic resonance spectroscopy (MRS) to muscle physiology is a new approach permitting a non-invasive examination of muscle performance. The technique can be used to measure multiple parameters in a single experimental protocol, including acetylcarnitine, phosphocreatine, IMCL, and maximum oxidative capacity (Qmax). In one study, multiple parameters were measured in the quadriceps of the left leg of thirteen Ironman volunteers and ten normal volunteers using MRS. The athletes had a higher IMCL content than the normal volunteers, as well as a higher Qmax and faster phosphocreatine resynthesis and recovery [57]. However, both LD isolation and MRS are currently unable to distinguish between intermyofibrillar and subsarcolemmal LDs.

## 7. Conclusions and Prospect

There are two distinct LD populations in skeletal muscle cells. Intermyofibrillar LDs are highly metabolically active, serving as an energy reservoir during acute exercise while subsarcolemmal LDs are fewer in number and are less active. T2DM patients accumulate lipids in large subsarcolemmal LDs. In contrast, athletes accumulate lipids in intermyofibrillar LDs, which are smaller and more numerous (Figure 2). The high surface area to volume ratio allows for more efficient and rapid liberation of their energy stores than those of the subsarcolemma.

It remains to be discovered how these two LD populations differ at the molecular level. It is possible that proteins specifically enriched in the intermyofibrillar LDs mediate a close association with mitochondria, facilitating energy use. There are likely other molecular differences that also explain why lipids can safely accumulate in intermyofibrillar LDs while diet-induced lipid storage in subsarcolemmal LDs leads to lipid toxicity and metabolic dysfunction. The mechanisms governing the differential storage of lipids in these two populations remain unknown. Further advances in spectroscopic, microscopic, and biochemical methods will be required to establish a deeper understanding of the molecular mechanisms underlying the athlete’s paradox.

## Figures and Tables

**Figure 1 cells-08-00249-f001:**
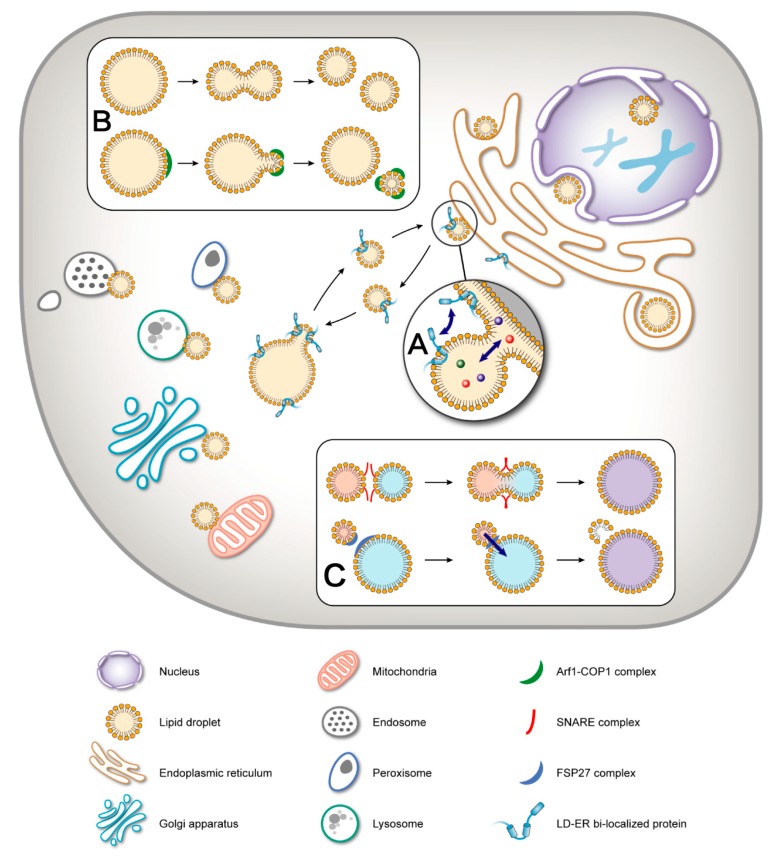
Dynamics of lipid droplets (LDs). As an organelle, a lipid droplet presents a very active status in cells, including movement, interaction with other cellular organelles, and size change. (**A**) LDs can interact with the endoplasmic reticulum (ER) through LD-ER contact sites. The action is to exchange material and information between the two organelles. (**B**) A large LD can be divided into smaller LDs by two means: fission and budding. (**C**) Two small LDs can form a large LD using fusion and neutral lipid transportation.

**Figure 2 cells-08-00249-f002:**
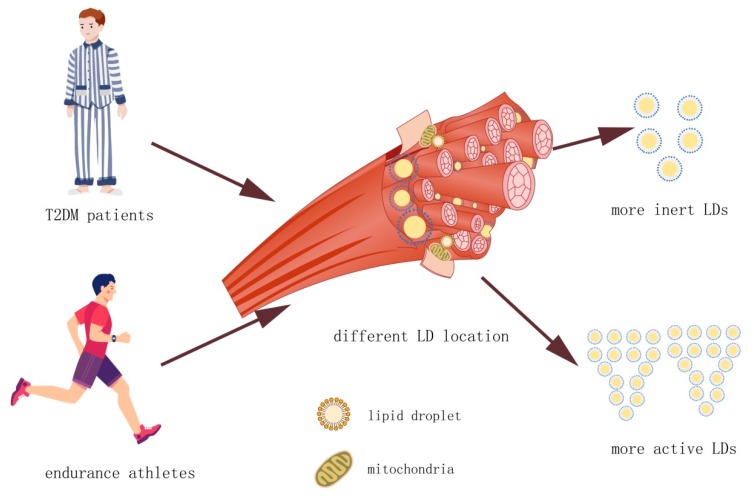
Size and location of LDs in skeletal muscle. Endurance athletes have the same amount of intermyofibrillar lipid in their skeletal muscle LDs as type 2 diabetes mellitus (T2DM) patients. However, their LDs are smaller in volume than the LDs in the skeletal muscle of T2DM and, therefore, the larger surface area provides higher lipolysis activity. The LDs of endurance athletes may also be contacted with mitochondria more than LDs in the skeletal muscle of T2DM, which provides the required energy for skeletal muscle with higher efficiency.

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
