# Peer review of "Skeletal Muscle Lipid Droplets and the Athlete’s Paradox"

_cells, 2019, doi:10.3390/cells8030249_

Round 1
Reviewer 1 Report
Li and colleagues have written an interesting review article on the so-called 'athlete's paradox'. This is a phenomenon where on one hand, the presence of lipid droplets (LD) in skeletal muscle positively correlates with insulin resistance in type 2 diabetes (T2D) patients, in athletes, the inverse situation pertains: the presence of LDs correlates positively with increased insulin sensitivity. This review dissects this paradox within the overall arena of T2D and notes that two distinct populations of LD exist within muscle fibres. Evidence is discussed that intramyofibrillar LDs are metabolically relevant in normal muscle whereas subsarcolemmal LDs are more associated with the pathobiology of T2D. The authors conclude and propose that new approaches are required to differentiate between these two distinct species of LD and hence their contribution to normal physiology versus T2D. A small number of corrections that will improve the manuscript are noted below.
1-The figures in the manuscript have only very minimal legends. The authors should add some additional narrative to expand the figure legends to make the Figures more accessible.
2-In several places, the authors refer to the ectopic storage of lipids (in non-adipose tissues). It would be useful to define, for a general readership whether such entities represent ectopic LDs per se, or otherwise other forms of deposit that differ in morphology and composition to LDs.
3-On pages 2 &3, the authors note that "To meet energy demand, IMCL can be hydrolyzed into FFAs, which are processed through β-oxidation in mitochondria to generate ATP and heat [9, 11]." It would be useful to clarify what the heat producing organ is in this context.
Author Response
Response to Reviewer 1 Comments
Comments and Suggestions for Authors
Li and colleagues have written an interesting review article on the so-called 'athlete's paradox'. This is a phenomenon where on one hand, the presence of lipid droplets (LD) in skeletal muscle positively correlates with insulin resistance in type 2 diabetes (T2D) patients, in athletes, the inverse situation pertains: the presence of LDs correlates positively with increased insulin sensitivity. This review dissects this paradox within the overall arena of T2D and notes that two distinct populations of LD exist within muscle fibres. Evidence is discussed that intramyofibrillar LDs are metabolically relevant in normal muscle whereas subsarcolemmal LDs are more associated with the pathobiology of T2D. The authors conclude and propose that new approaches are required to differentiate between these two distinct species of LD and hence their contribution to normal physiology versus T2D. A small number of corrections that will improve the manuscript are noted below.
1-The figures in the manuscript have only very minimal legends. The authors should add some additional narrative to expand the figure legends to make the Figures more accessible.
Response:
We thank the reviewer’s suggestion and revised the figure legend 1 and 2, including some detailed corrections to the contents of the two figures. The following is revised figure legends:
Figure 1. Dynamics of Lipid Droplets.
As an organelle, lipid droplet presents a very active status in cells, including movement, interaction with other cellular organelles, and size change. A LDs can be interacted with endoplasmic reticulum (ER) through LD-ER contact sites. The action is to exchange material and information between two organelles. B A large LD can be divided into small LDs by two means, fission and budding. C Two small LDs can form a large LD using fusion and neutral lipid transportation.
Figure 2. Sizes and Location of LDs in Skeletal Muscle.
Endurance athletes have the same amount of IMCL in their skeletal muscle LDs as T2DM patients. But their LDs are smaller in volume than LDs in the skeletal muscle of T2DM, and therefore larger in surface area that provides higher lipolysis activity. The LDs of endurance athletes are also contacted with mitochondria more than LDs in the skeletal muscle of T2DM, which provides the required energy for skeletal muscle with higher efficiency.
2-In several places, the authors refer to the ectopic storage of lipids (in non-adipose tissues). It would be useful to define, for a general readership whether such entities represent ectopic LDs per se, or otherwise other forms of deposit that differ in morphology and composition to LDs.
Response:
We thank the reviewer’s suggestion. We added detailed explanation of lipid ectopic storage in the revised manuscript, including excessive lipid storage in non-adipose tissues. The following is the revised text:
This in turn leads to the ectopic storage of neutral lipids in non-adipose tissues, such as skeletal muscle, liver and heart [3]. Two major human adipose tissues are described, white adipose tissue (WAT) and brown adipose tissue (BAT). WAT is mainly used to store energy, while BAT is mainly used to produce heat [4]. Ectopic lipid storage [5] refers to the lipids that cannot be consumed and stored in adipose tissues are over stored in non-adipose tissues, which, in turn, causes lipid metabolism disorder as well as lipid toxicity.
3-On pages 2 &3, the authors note that "To meet energy demand, IMCL can be hydrolyzed into FFAs, which are processed through β-oxidation in mitochondria to generate ATP and heat [9, 11]." It would be useful to clarify what the heat producing organ is in this context.
Response:
We thank the reviewer’s suggestion and revised the manuscript as following:
Lipids are stored as TAG in LDs within skeletal muscle cells, called intramyocellular lipid (IMCL) [11, 12]. To meet energy demand, IMCL can be hydrolyzed into FFAs, which are processed through β-oxidation in mitochondria to generate ATP and heat in skeletal muscle [11, 13].

Reviewer 2 Report
This review by Xuehan Li and colleagues attempts to discuss the current understanding of lipid droplets in skeletal muscle that may explain the Athletes Paradox – the situation where the amount of lipid (triglycerides) in skeletal muscle is equal in highly trained endurance athletes and type 2 diabetics, despite dramatic differences in insulin sensitivity.
Overall, this review is very superficial and fails to discuss critical areas of lipid droplet biology that have been explored in this area. For example, there is no discussion of lipid droplet proteins and how their abundance does or does not explain the Athlete’s Paradox. It also fails to discuss many key publications on this topic – PMIDs 27649157, 28560826, 28941236, 17510710, and 29443549 as examples.
Other issues:
1. Figure 1 adds very little to the manuscript and is very difficult to comprehend. It tries to highlight LD-organelle contacts/interactions yet the remainder of the manuscript does not discuss in any meaningful way the existing evidence for this to be a potential explanation linking LD biology to insulin sensitivity.
2. The entire discussion on Type 2 Diabetes is too broad and fails to present data on the existing literature on LD biology and insulin resistance in sufficient detail to be of use to the field.
3. The entire section on subcellular compartmentalization again fails to discuss the existing literature that has explored differences in subcellular features of LDs in models of obesity, insulin resistance or similar, especially lipolysis using dialysis probes and LD protein abundance. This is especially surprising as the most recent publication on the Athlete’s Paradox (PMID 30174227; Ref #41) reports differences in LD protein levels.
4. The section “New Approaches to the Study of Muscle Physiology” reads like a grant application, where the authors are proposing that their new technique will be able resolve various aspects of skeletal muscle LD biology and thereby identify differences that explain the Athlete’s Paradox. Further, very little insight is provided in the rest of the section on what underpins the Athelete’s Paradox.
Author Response
Response to Reviewer 2 Comments
Comments and Suggestions for Authors
This review by Xuehan Li and colleagues attempts to discuss the current understanding of lipid droplets in skeletal muscle that may explain the Athletes Paradox – the situation where the amount of lipid (triglycerides) in skeletal muscle is equal in highly trained endurance athletes and type 2 diabetics, despite dramatic differences in insulin sensitivity.
Overall, this review is very superficial and fails to discuss critical areas of lipid droplet biology that have been explored in this area. For example, there is no discussion of lipid droplet proteins and how their abundance does or does not explain the Athlete’s Paradox. It also fails to discuss many key publications on this topic – PMIDs 27649157, 28560826, 28941236, 17510710, and 29443549 as examples.
Response:
We thank the reviewer’s comments and suggestions. We have carefully analyzed these suggested literatures and included two of them in the revised manuscript as references 6 (27649157) and 36 (29443549) that discussed the relationship between athletes and LDs.
Other issues:
1. Figure 1 adds very little to the manuscript and is very difficult to comprehend. It tries to highlight LD-organelle contacts/interactions yet the remainder of the manuscript does not discuss in any meaningful way the existing evidence for this to be a potential explanation linking LD biology to insulin sensitivity.
Response:
We thank the reviewer’s suggestion and revised the figure legends 1 and 2 as following:
Figure 1. Dynamics of Lipid Droplets.
As an organelle, lipid droplet presents a very active status in cells, including movement, interaction with other cellular organelles, and size change. A LDs can be interacted with endoplasmic reticulum (ER) through LD-ER contact sites. The action is to exchange material and information between two organelles. B A large LD can be divided into small LDs by two means, fission and budding. C Two small LDs can form a large LD using fusion and neutral lipid transportation.
Figure 2. Sizes and Location of LDs in Skeletal Muscle.
Endurance athletes have the same amount of IMCL in their skeletal muscle LDs as T2DM patients. But their LDs are smaller in volume than LDs in the skeletal muscle of T2DM, and therefore larger in surface area that provides higher lipolysis activity. The LDs of endurance athletes are also contacted with mitochondria more than LDs in the skeletal muscle of T2DM, which provides the required energy for skeletal muscle with higher efficiency.
2. The entire discussion on Type 2 Diabetes is too broad and fails to present data on the existing literature on LD biology and insulin resistance in sufficient detail to be of use to the field.
Response:
We thank the reviewer’s comments and suggestions. We agree that this is important issue and we are going to summarize the relationship between LD biology and insulin resistance in more detail in a new review article. In this manuscript, we prefer to focus on the Athlete’s Paradox.
3. The entire section on subcellular compartmentalization again fails to discuss the existing literature that has explored differences in subcellular features of LDs in models of obesity, insulin resistance or similar, especially lipolysis using dialysis probes and LD protein abundance. This is especially surprising as the most recent publication on the Athlete’s Paradox (PMID 30174227; Ref #41) reports differences in LD protein levels.
Response:
We thank the reviewer’s comments. In fact, we did cite this literature (PMID 30174227) several times. We think the current reported evidence for IMCL and LD surface proteins is not sufficient to explain the Athlete’s Paradox. Therefore, we paid more attention to the distribution of LDs in the skeletal muscles before and after exercise.
4. The section “New Approaches to the Study of Muscle Physiology” reads like a grant application, where the authors are proposing that their new technique will be able resolve various aspects of skeletal muscle LD biology and thereby identify differences that explain the Athlete’s Paradox. Further, very little insight is provided in the rest of the section on what underpins the Athelete’s Paradox.
Response:
We thank the reviewer’s comments. Indeed, our laboratory has made some contributions to the purification of skeletal muscle LDs, and we have also obtained some results that demonstrate the tight contact between LDs and mitochondria. These findings may provide a clue to solve the Athletes’ Paradox. We also believe that more information can be obtained using proteomic analysis of LDs isolated from the muscles of endurance athletes and T2DM.

Reviewer 3 Report
In the current review article, the authors Li et al. explain the Athlete’s paradox: trained athletes accumulate lipids in their skeletal muscle, and size of lipid droplets positively correlate with insulin sensitivity. Lipid droplets have been on the other hand suggested to be involved in type 2 diabetes, atherosclerosis, steatosis, and obesity, and the authors discuss actually in large part of the manuscript about the pathological effects of lipid droplets. The authors then propose a reason of the paradox that subcellular compartmentalization (intermyofibrillar or subsarcolemmal) of lipid droplets and the mitochondria make difference whether lipid droplets have physiological or pathological effect. This is a highly interesting and important point. Because it might be possible to create new therapeutic strategy for patients with type 2 diabetes, atherosclerosis, steatosis, and obesity, if background molecular mechanisms of lipid droplet accumulation in different subcellular compartmentalization are identified and characterized. The authors however say that it remains to be discovered how two lipid droplet populations differ at the molecular level.
1. The authors propose there is a close association between lipid droplets and mitochondria in their manuscript. There are actually several publications supporting the idea: Ferreira et al. Proteomics 2010;10:3142 analyzed intermyofibrillar and subsarcolemmal mitochondria in skeletal muscle and performed mass spectrometric analyses. Ritov et al. Diabetes 2005;54:8 suggests deficiency of subsarcolemmal mitochondria in obesity and muscle insulin resistance. A recent publication Benador et al. Cell Metabolism 2018;27:869 suggests that mitochondria bound to lipid droplets have unique bioenergetics, composition, and dynamics supporting lipid droplet expansion. The authors in the current manuscript are requested to recheck several publications and deeper discuss about (potential) molecular explanation.
Author Response
Response to Reviewer 3 Comments
Comments and Suggestions for Authors
In the current review article, the authors Li et al. explain the Athlete’s paradox: trained athletes accumulate lipids in their skeletal muscle, and size of lipid droplets positively correlate with insulin sensitivity. Lipid droplets have been on the other hand suggested to be involved in type 2 diabetes, atherosclerosis, steatosis, and obesity, and the authors discuss actually in large part of the manuscript about the pathological effects of lipid droplets. The authors then propose a reason of the paradox that subcellular compartmentalization (intermyofibrillar or subsarcolemmal) of lipid droplets and the mitochondria make difference whether lipid droplets have physiological or pathological effect. This is a highly interesting and important point. Because it might be possible to create new therapeutic strategy for patients with type 2 diabetes, atherosclerosis, steatosis, and obesity, if background molecular mechanisms of lipid droplet accumulation in different subcellular compartmentalization are identified and characterized. The authors however say that it remains to be discovered how two lipid droplet populations differ at the molecular level.
1. The authors propose there is a close association between lipid droplets and mitochondria in their manuscript. There are actually several publications supporting the idea: Ferreira et al. Proteomics 2010;10:3142 analyzed intermyofibrillar and subsarcolemmal mitochondria in skeletal muscle and performed mass spectrometric analyses. Ritov et al. Diabetes 2005;54:8 suggests deficiency of subsarcolemmal mitochondria in obesity and muscle insulin resistance. A recent publication Benador et al. Cell Metabolism 2018;27:869 suggests that mitochondria bound to lipid droplets have unique bioenergetics, composition, and dynamics supporting lipid droplet expansion. The authors in the current manuscript are requested to recheck several publications and deeper discuss about (potential) molecular explanation.
Response:
We thank the reviewer’s comments and suggestions. We have described the relationship between LDs and mitochondria in more detail according to your suggestion, and included the references. The following is revised part:
Lipid Droplets and Mitochondria
LDs interact with mitochondria, and the distribution of LDs and mitochondria in skeletal muscle type I muscle fibers have been observed by traditional light microscope and laser confocal three-dimensional reconstruction technique. It is found that LDs are mainly distributed in the aggregation of mitochondria [47]. Meanwhile, peridroplet mitochondria (PDM) in brown adipocytes support LD expansion because Perilipin5 induces mitochondrial recruitment to LDs and increases the synthesis of triacylglycerol (TAG) dependent on ATP synthase [48].
It has also been reported that the protein content of each intermyofibrillar mitochondria in rat skeletal muscle is nearly twice as high as that of Subsarcolemmal mitochondria, which means that intermyofibrillar mitochondria have higher activity [49]. At the protein level, SNAP23 is one of the proteins found to regulate the interaction between LDs and mitochondria [50]. In skeletal muscle, SNAP23 is partially localized on the cell membrane and involved in the translocation of insulin-sensitive glucose transporter 4 (GLUT4) to the cell membrane [51]. In fatty acid treated cells, with the increased LDs, SNAP23 will be more localized on the surface of LDs, which enhances the interaction between LDs and mitochondria, and reduces plasma membrane GLUT4 that, in turn, decreases glucose uptake [51].
Multiple lines of evidence have established that LDs can physically associate with mitochondria, suggesting a functional link in the mobilization and use of energy reserves [40]. In addition to the accumulation of intramyofibrillar LDs, long-term endurance training also increases the biogenesis of mitochondria [52]. There is the evidence that mitochondrial dysfunction can lead to insulin resistance [53]. The activity of the electron transfer chain in the submembranous mitochondria of patients with type 2 diabetes and obesity is significantly lower than that of thin volunteers, suggesting that mitochondrial dysfunction may also lead to type 2 diabetes. Therefore, a more detailed understanding of the role of IMCL in health and disease will require the techniques of cell biology and biochemistry.

Reviewer 4 Report
“Athlete’s paradox” in which endurance-trained athletes, who possess a high oxidative capacity and enhanced insulin sensitivity, also have higher intramyocellular lipid was described for the first time almost two decades ago. The molecular mechanisms leading to athlete’s paradox” are still poorly understood, bud the large body of evidence suggests the importance of different Lipid Droplets (LD) morphology and localization in skeletal muscles of endurance-trained athletes and the general population. The review paper by Li and co-workers discussing current knowledge about athlete’s paradox is the first one summarizing this phenomenon. The manuscript is well written and easy to follow. It will be an important contribution to enhance general knowledge, especially important in the era of worldwide lipid metabolic diseases epidemy. However, already known details concerning LD biochemistry and morphology are omitted from the paper. The contribution of LD morphology, subcellular localization, and differences in protein content to the athlete's paradox was for example subject to recent studies by Daemen and co-workers (2018 Mol Metab) cited in the current review. Both figures could be also improved. In the Fig1 diagrams showing fission, fusion and biogenesis should be labeled A, B, C. To the fig 2 association of the intermyofibrillar LD with mitochondria should be emphasized.
Author Response
Response to Reviewer 4 Comments
Comments and Suggestions for Authors
“Athlete’s paradox” in which endurance-trained athletes, who possess a high oxidative capacity and enhanced insulin sensitivity, also have higher intramyocellular lipid was described for the first time almost two decades ago. The molecular mechanisms leading to athlete’s paradox” are still poorly understood, bud the large body of evidence suggests the importance of different Lipid Droplets (LD) morphology and localization in skeletal muscles of endurance-trained athletes and the general population. The review paper by Li and co-workers discussing current knowledge about athlete’s paradox is the first one summarizing this phenomenon. The manuscript is well written and easy to follow. It will be an important contribution to enhance general knowledge, especially important in the era of worldwide lipid metabolic diseases epidemy. However, already known details concerning LD biochemistry and morphology are omitted from the paper. The contribution of LD morphology, subcellular localization, and differences in protein content to the athlete's paradox was for example subject to recent studies by Daemen and co-workers (2018 Mol Metab) cited in the current review. Both figures could be also improved. In the Fig1 diagrams showing fission, fusion and biogenesis should be labeled A, B, C. To the fig 2 association of the intermyofibrillar LD with mitochondria should be emphasized.
Response:
We thank the reviewer’s positive comments and useful suggestions. We agree that LD proteome and lipidome play important roles in its functions in Athlete’s Paradox, and suggested in this paper to isolate LDs from athletes and T2DM and analyze the organelles biochemically and morphologically. In fact, we had a review previously to summarize the correlation between LD proteins and metabolic diseases. Here, we would like to focus on the relationship between Athlete’s Paradox and LD size and distributions. Based on the reviewer’s suggestion, we made a couple of changes to the contents of the two figures, as well as included detailed figure legends in the revised manuscript. The following are the changes:
Figure 1. Dynamics of Lipid Droplets.
As an organelle, lipid droplet presents a very active status in cells, including movement, interaction with other cellular organelles, and size change. A LDs can be interacted with endoplasmic reticulum (ER) through LD-ER contact sites. The action is to exchange material and information between two organelles. B A large LD can be divided into small LDs by two means, fission and budding. C Two small LDs can form a large LD using fusion and neutral lipid transportation.
Figure 2. Sizes and Location of LDs in Skeletal Muscle.
Endurance athletes have the same amount of IMCL in their skeletal muscle LDs as T2DM patients. But their LDs are smaller in volume than LDs in the skeletal muscle of T2DM, and therefore larger in surface area that provides higher lipolysis activity. The LDs of endurance athletes are also contacted with mitochondria more than LDs in the skeletal muscle of T2DM, which provides the required energy for skeletal muscle with higher efficiency.

Round 2
Reviewer 3 Report
The authors Li et al. revised thier current manuscript about skeletal muscle lipid droplets and the athelete's paradox. Most importantly the authors discuss further lipid droplets and mitochondria and refer several key findings in addition. The reviewer does not see any critical concern in the revised version.